# OpenReview forum: "Steering Beyond the Support: Adversarial Training on Unsupervised Jailbroken Activation Simulation"
_ICML.cc/2026/Conference — ICML 2026 regular_

### Official Review · Reviewer_CizA · 2026-03-10

**Soundness:** 3
**Presentation:** 3
**Significance:** 3
**Originality:** 3
**Overall Recommendation:** 4
**Confidence:** 2

**Summary:**

The paper studies robust defense against jailbreak prompts in aligned LLMs using activation steering. Existing safety steering methods rely on supervised jailbreak datasets, which limits generalization to unseen jailbreak attacks. The authors propose a bi-level adversarial training framework that generates simulated jailbreak activations via unsupervised latent direction discovery, and trains a potential-function-based steering field to push harmful activations toward refusal regions while leaving benign activations unchanged. Experiments on three aligned LLMs (LLaMA-3-8B, Mistral-V2-7B, Qwen-2.5-7B) and six jailbreak attacks show improved jailbreak robustness while maintaining utility.

**Compliance With Llm Reviewing Policy:**

Affirmed.

**Final Justification:**

My concerns have been addressed, so I am maintaining my positive score.

**Key Questions For Authors:**

The key questions are as follows:

1. Coverage of Latent Directions vs. Real Jailbreak Manifolds. The proposed method relies on latent directions discovered from refusal activations to simulate jailbreak activations. However, it is unclear whether these directions adequately cover the activation regions induced by real jailbreak prompts. Could the authors provide stronger evidence that the discovered latent directions meaningfully approximate real jailbreak activation manifolds (e.g., through additional representation similarity analysis, coverage metrics, or experiments with previously unseen jailbreak strategies)?

2. Evaluation Under Truly Unseen Jailbreak Attacks. The paper claims improved generalization beyond the training support, yet the evaluation appears limited to several known jailbreak families. Have the authors evaluated the method on newer or structurally different jailbreak strategies, or attacks that were not used during any stage of the method development? Demonstrating robustness on genuinely unseen attack paradigms would help substantiate the generalization claim and could improve my confidence in the practical effectiveness of the approach.

3. Inference Overhead and Deployment Cost. The framework involves activation manipulation and gradient-based updates during inference. Could the authors clarify the runtime overhead, latency impact, and computational cost compared with standard decoding? Additionally, how does the method scale to long responses or multi-turn dialogue scenarios? Providing quantitative overhead measurements would help assess whether the method is practical for real-world deployment.

4. Stability of Steering During Long Generation. Since activation steering modifies intermediate representations, there may be risks of instability or drift during long generation sequences. Have the authors evaluated whether the steering mechanism remains stable over long responses or iterative decoding steps? If the method introduces instability, mitigation strategies would be helpful to discuss.

5. Interaction with Alignment Methods. Modern aligned LLMs already include RLHF, safety fine-tuning, and refusal policies. How does the proposed steering mechanism interact with these existing alignment methods? For example, does it consistently improve robustness across models with different alignment pipelines, or could there be cases where it conflicts with existing safety policies?

**Limitations:**

The paper discusses several practical considerations but could provide a more explicit discussion of limitations and potential risks:

1.The method relies on the assumption that latent directions discovered from refusal activations can approximate the representation regions associated with jailbreak prompts. This assumption may not always hold, especially for sophisticated jailbreak strategies that exploit reasoning chains, multi-turn interactions, or context manipulation. A clearer discussion of this assumption and its potential failure modes would improve transparency.

2.The evaluation focuses primarily on a set of known jailbreak attack families. As jailbreak strategies evolve rapidly, robustness demonstrated on existing benchmarks may not fully reflect performance against future attack paradigms. The authors could strengthen the limitations section by acknowledging this gap and discussing potential ways to extend the method to adapt to new attack patterns.

3.The proposed steering mechanism introduces additional inference-time computation and activation manipulation, which may raise practical concerns regarding latency, scalability, and stability during long generations or multi-turn dialogue. Explicitly discussing these deployment considerations would help readers better understand the trade-offs of the method.

From a societal perspective, techniques that manipulate internal model representations may also potentially be repurposed to steer models toward harmful behaviors if misused. A brief discussion of safeguards or responsible deployment considerations would further strengthen the paper’s treatment of broader impacts.

**Strengths And Weaknesses:**

Strength:

1.Clear Motivation. The paper targets a real limitation of current LLM safety steering methods: dependence on a fixed jailbreak dataset. The key observation is that supervised steering methods fail to generalize to out-of-distribution jailbreak prompts, which is an important challenge for real-world deployment. This motivation is well articulated and empirically supported.

2.Novel Perspective. Instead of collecting more jailbreak prompts, the paper proposes simulating jailbreak activations in representation space via latent directions. This conceptual shift is a meaningful contribution.

3.Mechanistic Analysis. The paper includes mechanistic interpretation like visualization of simulated jailbreak activation expansion, analysis of subspace coverage and correlation between coverage and robustness. These analyses strengthen the argument that the method works for the intended reason.

4.Comprehensive Evaluation. The evaluation includes 3 models, 6 jailbreak attacks, robustness metrics and capability preservation, over-refusal analysis and ablation studies.

Weakness:

1.Strong Reliance on the Latent Direction Discovery Assumption. The entire framework assumes that latent directions discovered from refusal activations can approximate unseen jailbreak activation regions. However, there is no theoretical guarantee that these directions cover real jailbreak manifolds and the method implicitly assumes that jailbreak behaviors lie near refusal activations in latent space. This assumption may not hold for sophisticated jailbreaks.

2.Potential Overclaim. The paper repeatedly claims improved generalization beyond training support. However, evaluation only includes known jailbreak families and simulated activations are validated using the same benchmark attacks. Thus the claim of generalization may be overstated, as true unseen jailbreak strategies are not tested.

3.Practical Deployment Questions. The method introduces several practical concerns, like multiple gradient ascent steps during inference, additional neural network or activation manipulation inside the model. And the paper does not analyze latency overhead, stability of steering in long generations and even robustness under multi-turn dialogue. These issues matter for real-world deployment.

---

> ### Author Rebuttal · Authors · 2026-03-29
>
> ## (W1, Q1, L1) On the local proximity assumption between refusal harmful activations and jailbreak activations
>
> We thank the reviewer for highlighting this important assumption. From empirical observation, we find that the latent directions discovered via unsupervised simulation not only induce diverse jailbreaking behaviors that are similar to different prompt jailbreaks   such as multilingual, hypothetical senario, role playing (Appendix I), but also significantly expand coverage over real jailbreak activation subspaces (Figure 6, 7), and Figure 4,5 qualitatively explains it via TSNE visualization. This suggests that refusal-state harmful activations provide a meaningful anchor point for exploring nearby failure regions.
>
> We agree that this assumption does not provide a formal guarantee of exhaustively covering all jailbreak modes. As discussed in the limitations (Appendix A), jailbreak behaviors arising from highly non-natural inputs (e.g., encoded or ciphered prompts) or complex reasoning trajectories may correspond to activation regions that are weakly reachable via linear latent extrapolation. This points to a broader limitation shared by steering-based defense approaches: robustness is bounded by the expressive capacity of the activation space.
>
> Future work may explore richer jailbreak simulation strategies, such as non-linear latent transformations, step-wise gradient search in activation space, or compositional use of multiple latent directions, to further expand coverage of complex jailbreak behaviors.
>
> ---
>
> ## (W2, Q2, L2) Evaluation on real unseen jailbreaks
>
> Our setting is explicitly designed as **zero-shot jailbreak defense**, meaning that no labeled jailbreak prompts are used during training signal construction. Instead, simulated jailbreak activations are generated from refusal-state harmful requests via unsupervised latent direction discovery. Therefore, we do perform real unseen jailbreak testing in zero-shot setting.  Results show that activation-space extrapolation from refusal states can provide meaningful protection against jailbreak strategies.
>
> ---
>
> ## (W3, Q3, Q4, L3) Inference cost and generation stability
> ### Runtime overhead
> We clarify that steering is applied only once during the **prefill stage**, after the model completes the forward pass over input tokens and before KV cache reuse for autoregressive decoding.
> $$\tilde{h}_l = h_l + \lambda r,$$
> where $r$ is obtained via a small number $K$ of gradient steps on the learned potential.
>
> As analyzed in Appendix E, this introduces negligible runtime overhead compared to standard decoding, the time complexity is O(KP), K is the number of gradient step, P is the number of parameters for potential function f, the time complexity is irrelevent to the sequence length, since $f$ proceesses the hidden state of a single token position from a single layer. In our implementation, the additional latency is approximately **0.08 seconds for a single input with arbitary length**, and **0.34 seconds for batch size 10**, indicating good scalability for practical deployment.
>
> ### Stability during long generation
>
> Regarding stability, no prior arts that directly benchmarks long multi-turn dialogue stability for safety steering. However, closely related activation-steering literature provides supporting evidence that representation-level interventions can stably alter full-sequence behavior with limited side effects on other capabilities. For example, refusal-direction paper shows that adding/removing a single direction can reliably induce or suppress refusal over complete generations while having minimal effect on benign capabilities. In our case, we also use single direction addition as the standard steering method, but activation addition comes from gradient steps on the potential function $f$. Thus, while a long-dialogue benchmark would be valuable future work, the mechanism and prior evidence both suggest that severe long-horizon instability is unlikely.
>
> ---
>
> ## Q5 Interaction with alignment methods
>
> Safety steering should be viewed as **complementary to alignment techniques such as RLHF**, rather than as a replacement. Alignment methods shape the model’s global safety during training, whereas steering provides a test-time mechanism for dynamically adjusting model behavior in response to potentially adversarial inputs.
>
> In particular, steering allows safety intervention strength to scale with attack severity (e.g., via more gradient steps or larger update magnitude), which can be especially useful when weakly aligned models encounter strong jailbreak attempts.
>
> Safety policy could be instantiated via some external guard models (e.g., llama guard family). However, such guard models themselves also need to be trained on known jailbreaks, and still vulnerable to other jailbreaks. By contrast, steering directly modifies internal representations, providing an additional layer of defense that operates within the model’s generation dynamics.

---

> > ### Author Rebuttal · Reviewer_CizA · 2026-04-03
> >
> > The author preliminarily address my concerns.

---

> > > ### Author Response · Authors · 2026-04-03
> > >
> > > We are glad that our response preliminarily address your concerns, and thank you again for the thoughtful and constructive feedback to make our work more solid, all the best!

---

### Official Review · Reviewer_23oj · 2026-03-11

**Soundness:** 3
**Presentation:** 3
**Significance:** 3
**Originality:** 3
**Overall Recommendation:** 4
**Confidence:** 3

**Summary:**

This paper aim to tackle the failure on unseen jailbreaks problem base on unsupervised latent direction discovery. It propose a bi-level adversarial training framework for zero-shot jailbreak defense by simulating diverse jailbroken activations by extrapolating from refusalstate harmful-request activations via unsupervised latent direction discovery.

**Compliance With Llm Reviewing Policy:**

Affirmed.

**Final Justification:**

My final opinion is accept.

**Key Questions For Authors:**

Same questions with weakness. However, I still look forward to comments from other reviewers and valuable feedbacks from authors for improving the initial scores.

**Limitations:**

Yes

**Strengths And Weaknesses:**

Strength:
1.It has extensive quantitative and quality analysis with clear opinion.
2.It is well-written with clear motivation and explanation of methodology details.

Weakness：
1.The experimental section is promising but still somewhat limited for a strong generalization claim. In particular, evaluation on more challenging OOD settings (e.g., multilingual, encoded, or highly obfuscated jailbreaks), along with clearer baseline matching and stronger sensitivity analysis, would make the conclusions more convincing.
2.The methodological justification is currently weaker than the empirical claims. Although the OT-based potential field and bi-level adversarial training are interesting, the paper does not clearly establish why these components should generalize better in principle, nor does it sufficiently disentangle which part of the framework contributes most to the gains.

---

> ### Author Rebuttal · Authors · 2026-03-29
>
> ## W1 Generalization under OOD Settings
>
> We thank the reviewer for highlighting the need for stronger OOD evaluation. We agree that our current experiments do not fully cover challenging distribution shifts such as multilingual or encoded (cipher) jailbreaks.
>
> As discussed in limitations (Appendix A), latent direction discovery is constrained by its expressiveness and may not fully capture extreme language transformations (e.g., cipher). To further evaluate this, we conduct additional experiments on Mistral-7B-v0.2 under multilingual and cipher jailbreak settings. These attacks are **not seen during training for any method**.
>
> |Method|Multilingual↓|Cipher↓|OR-FPR↓|
> |-|-|-|-|
> |Base |47.45|20.38|1.7|
> |LAT|12.45|16.64|32.3|
> |ROSI|15.54|17.80|35.7|
> |AlphaSteer|10.89|**16.05**|25.6|
> |Ours|**3.47**|16.33|**20.7**|
>
> Our method shows stronger robustness on multilingual attacks, since latent direction discovery can capture cross-lingual jailbreak behaviors (Appendix I, Franch/Chinese/German). However, all methods struggle on cipher attacks, indicating a limitation under extreme OOD. This shows that our contribution lies in **activation-space generalization**, rather than fully solving all input-level transformations.
> ## W1 Clear baseline matching
> We kindly refer to **Q2 W3 Q4 Setting details across methods** response for **1st reviewer uzz9**.
> ## W1 Hyperparameter Sensitivity Analysis
> We analyze sensitivity to **latent magnitude $R$** and **number of directions $|V|$** (steering steps are discussed in Appendix H).
> ### (1) Magnitude $R$
> We select $R$ based on the **linear explainability ratio** (LER, Appendix D), measuring how well a local linear direction explains activation changes. We consider three regimes:
>
> | LER |Interpretation|Effect|avg.SR|
> |-|-|-|-|
> |0.25~0.5|Highly linear regime|$R$ too small → weak intervention|0.11|
> |0.5~0.75|Moderate nonlinearity|$R$ moderate  → effective jailbroken simulation|0.56|
> |0.75~1.0|Strong nonlinearity|$R$ large but inaccurate → large intervention but gibberish outputs|0.24|
>
> In practice, (0.5-0.75) achieves the best trade-off, simulating effective jailbreaks, while too small or too large $R$ both degrade simulation effectiveness.
>
> ### (2) Number of directions $|V|$
> We evaluate $|V|$ using the number of effective jailbroken directions (SR > 50)
>
> |$\|V\|$| # Effective directions |
> |-|-|
> |64|48|
> |128|86|
> |256|135|
> |512|207|
> |1024|294|
>
> We observe that increasing $|V|$ improves performance initially, but the number of effective directions grows sublinearly, and gains saturate beyond 512. Also, larger $\|V\|$ requires more GPU VRAM and slow down simulation process.
> Overall, good performance is achieved with: R under moderate linear explaination ratio, and sufficiently large but not excessive $\|V\|$
>
> ## W2 Methodological Justification and Ablation
> We clarify the roles of (i) OT-based potential and (ii) bi-level adversarial training.
>
> ### (1) OT-based potential
> Recall steering performs activation addition as: $$\tilde{h}= h + \lambda r$$
> Activation $h\in \mathbb{R}^d$.
>
> Our OT potential formulation enables arbitarily complex steering function for enough expressiveness, and supports multi-step steering (gradient ascent). Baseline method AlphaSteer uses linear steering function: $r = Wh$, and this is equivalent as a special case in our framework:  $f_\phi =\frac{1}{2}  h^\top \Phi h$, $\Phi \in \mathbb{R}^{d\times d}$, since $\nabla f(h) = \Phi h$. And the number of gradient update step K=1. This formulation is particularly suitable for highly heterogeneous, multi-mode jailbreak distributions. We will include the math proof during revision. We add experiment on **Mistral-7b-v0.2**, measuring robustness gain from this formulation.  Here $\eta=0.2$ and $K=20$
>
> |Method|Parameterization|Gradient Update|GCG↓|AutoDAN↓|GPTFuzz↓|PAIR↓|TAP↓|FewShot↓|OR-FPR↓|
> |-|-|-|-|-|-|-|-|-|-|
> |Linear-1 Step (AlphaSteer)|$f_\phi(h)=\frac{1}{2}  h^\top \Phi h$|1-step, lr=$\eta K$|8.11|1.95|3.69|7.75|7.98|7.54|23.76|
> |Linear-K Step|$f_\phi(h)= \frac{1}{2} h^\top \Phi h$|K-step, lr=$\eta$|7.45|1.54 |3.42|6.67|7.03|7.02|22.45|
> |MLP-K Step (Ours)|$f_\phi(h) = \text{mlp}(h)$|K-step, lr=$\eta$|**7.10**|**1.35**|**3.21**|**6.31**|**6.69**|**6.92**|**20.74**|
>
> Using mlp as steering function and multi step update yields better effectiveness, supporting the OT potential design.
>
> ### (2) Bi-level training
> The inner step continuously discovers **hard latent directions** that induce jailbreak behavior, while the outer step learns to steer them. This process progressively expands the **coverage of adversarial activation regions**, analogous to adversarial training. Also, we have **Empirical validations in Sec. 9.3** : When removing the adversarial training, and using only fixed latent directions, we observe a clear drop in robustness due to insufficient subspace coverage on real jailbreak distribution. This indicates that the gains critically depend on the **adaptive discovery of hard cases and coverage expansion**.

---

> > ### Author Rebuttal · Reviewer_23oj · 2026-04-01
> >
> > Thanks for your explanation. My concerns have been addressed.

---

> > > ### Author Response · Authors · 2026-04-01
> > >
> > > Thank you for raising score for our work, all the best!

---

### Official Review · Reviewer_uZZ9 · 2026-03-17

**Soundness:** 2
**Presentation:** 2
**Significance:** 3
**Originality:** 3
**Overall Recommendation:** 4
**Confidence:** 4

**Summary:**

This paper studies activation steering for jailbreak defense under unseen or out-of-distribution attacks. Instead of learning steering only from a fixed supervised jailbreak set, the paper proposes to simulate diverse jailbroken activations through unsupervised latent direction discovery, and then train a potential-induced steering field with a bi-level adversarial objective. The method is evaluated on three aligned LLMs and six jailbreak families, and the paper reports improved robustness together with increasing subspace coverage during training.

**Compliance With Llm Reviewing Policy:**

Affirmed.

**Final Justification:**

I have raised my score.

**Key Questions For Authors:**

1. The paper argues that the method generalizes beyond a fixed supervised jailbreak support. Could the authors provide stronger evidence for this claim under more challenging settings, for example adaptive white-box attacks against the learned steering field?
2. For the comparisons in Table 3 and Table 4, are all methods trained and evaluated under closely matched conditions, including the same data budget, the same amount of tuning effort, and the same evaluation pipeline? This would make it easier to judge how much of the gain comes from the proposed framework itself.
3. The subspace-coverage analysis in Section 9 is interesting, but how sensitive is it to choices such as the PCA rank r, the number of selected simulated points K, and the definition of attack-family subspaces?
4. The method contains several interacting design choices, including the latent direction magnitude R, the number of discovered directions, the number of steering steps, and the potential objective. Could the authors clarify which of these are most important in practice?

**Limitations:**

Yes

**Strengths And Weaknesses:**

**Strengths**
- The paper targets an important problem in LLM safety, namely whether activation steering can generalize beyond a fixed supervised jailbreak set.
- The overall method is reasonably well motivated. In particular, combining unsupervised jailbreak activation simulation with a learned steering field is an interesting approach to unseen attacks.
- The reported empirical results are encouraging. Across several models and attack families, the method is competitive with or better than multiple steering-based baselines.

**Weaknesses**
- The main generalization claim is somewhat stronger than what the current evidence fully supports. The evaluation still focuses on a fixed collection of standard jailbreak families, without stronger adaptive or white-box attack evidence.
- Some of the interpretation remains fairly heuristic. In particular, the subspace-coverage analysis is interesting, but it is still unclear how directly this proxy explains robustness across models and attack types.
- More detail would help in assessing the experimental setup, especially how sensitive the final performance is to the simulation and steering design choices.
- The writing should be improved further.

---

> ### Author Rebuttal · Authors · 2026-03-29
>
> We thank reviewer for constructive and thoughtful feedback, we address concerns as below.
>
> ## W1,Q1 Claim about generalization
> - The paper argues that the method generalizes beyond a fixed supervised jailbreak support.
> - Robustness of white-box adaptive attack against the learned steering field.
> ## Response
> - We clarify that our method as  **zero-shot jailbreak defense from direct harmful queries**,rather than generalizes beyond a fixed supervised jailbreak support. In 6.1 Experimental Setup, we detailed the training data comes from AdvBench and OR-Bench Toxic, which are direct harmful queries, then test the learned steer on jailbreaks.
> - We designed two adaptive attack settings: attacker optimze adversarial input (using GCG attack) against model under steering protection, with (1) and without (2) acess to the steering potential parameter.
>
> We have GCG standard loss $L_{CE}$ for (1) and additional adaptive loss $L_{CE}$ + $ L_{adv}$ for (2), to be hard to steer.
>
> $L_{CE} = \min_{s \in \mathcal{V}^m}-\sum_{t=1}^{T}\log p(y_t^{\mathrm{harm}} \mid x \oplus s, y_{<t}^{\mathrm{harm}};\theta,\phi).$
>
> $L_{adv} = \min_{s \in \mathcal{V}^m} \|\nabla f_{\phi}(h_\ell(x \oplus s;\theta,\phi))\|_2^2$
>
> |Method|SR↓|Final Loss↑|
> |-|-|-|
> |Base|63.34|0.33|
> |Base+Steer|7.10|NA|
> |Adaptive (1)|12.55|0.96|
> |Adaptive (2)|15.54|0.82|
>
> We observe our method remain certain robustness under adaptive attacks.
>
> loss curve is here: [link](https://anonymous.4open.science/r/icml_rebuttal-F814/ada_loss_curves.png)
>
> ## W2,Q3  Subspace coverage mertic explaination
> - How subspace-coverage analysis explains robustness gain is unclear
> -  Explain the choices for PCA rank r, the number of selected simulated points K, and the definition of attack-family subspaces
> ## Response
> - **What subspace coverage measures.**
> it quantifies whether the **simulated jailbreak activations effectively cover the real jailbreak activation manifold**. Concretely, for each jailbreak family, we first estimate a low-dimensional attack subspace via PCA over real jailbreak activations. We then project all simulated activations onto this subspace and compute their projection magnitudes. A representative subset of the **top-K simulated activations (we use K = 20)** — i.e., those with the largest projection values — is selected, and their average projection strength is used to define the reported coverage score. This  ensures that the metric reflects **coverage of the relevant jailbreak manifold**.
>
> **Why increased coverage has a causal connection to robustness.**
> During training, we maximize the potential gradient norm at simulated jailbreak activations. As a result, when simulated activations increasingly cover the real jailbreak manifold, the learned steering potential develops **stronger gradients in neighborhoods of real jailbreak activations**, making it easier to transport them into refusal regions in limited steps.
>
> - **Rationale behind sensitivity-related design choices.**
> The parameters mentioned by the reviewer are primarily chosen to ensure **causal faithfulness and stability of manifold estimation**, rather than for performance tuning.
>
> **Definition of attack subspaces** via PCA over jailbreak-family activations provides a practical and relatively stable approximation of the underlying jailbreak activation manifold.
>
> **PCA rank \(r = 10\)** is selected as a compromise between overly low ranks (e.g., \(r < 3\)), which fail to capture the geometric variability of jailbreak activations, and overly high ranks (e.g., \(r > 100\)), which begin to model noise and reduce interpretability.
>
> **Number of selected simulated activations \(K = 20\)** is chosen to obtain a sufficiently robust estimate of coverage near the jailbreak manifold. Using too few points (e.g., \(K < 5\)) leads to unstable estimates, while using all simulated activations would dilute the metric by including points that are not relevant to the current jailbreak manifold.
>
>
> ## Q2 W3  Q4 Setting details across methods, hyper-parameter choices.
> - How sensitive the final performance is to the simulation and steering design choices, latent direction magnitude R, the number of discovered directions, the number of steering steps?
>
> - For the comparisons in Table 3 and Table 4, are all methods trained and evaluated under closely matched conditions?
>
> ## Response
> -  For sensitivity analysis, we kindly refer this concern to the response for **the 2nd reviewer 23oj W1 Hyperparameter Sensitivity Analysis**
>
> - We clarify that all methods are trained and evaluated under closely matched conditions, and we will further emphasize this in the revision.
>
> **Training data:** All methods use the same harmful and benign datasets for training.
> **Training:** We follow the official implementations for each baseline.
> **Evaluation:** All methods are evaluated using the same pipeline (including the same StrongREJECT scoring and attack/benign/safety boundary datasets).

---

> > ### Author Rebuttal · Reviewer_uZZ9 · 2026-04-03
> >
> > The rebuttal helps narrow the intended claim, but in doing so it also reinforces my original concern that the paper’s broader generalization framing is currently stronger than what the evidence fully supports. The added adaptive evaluation is a useful step, but still seems limited relative to the strength of the claim. The explanation of the subspace coverage metric is also clearer, yet the argument connecting coverage to robustness remains largely heuristic, and the response does not fully address my questions about sensitivity to the main design choices. I also still would have preferred more explicit evidence that all methods are compared under closely matched budgets and tuning conditions.
> >
> > Overall, the rebuttal improves clarity, but it does not substantially change my assessment, so I will keep my original score.

---

> > > ### Author Response · Authors · 2026-04-03
> > >
> > > Thank you very much for the thoughtful and constructive feedback.
> > >
> > > We would like to clarify the intended scope of our claim. Our primary goal is **not to establish a universal strong generalization guarantee across arbitrary jailbreak attacks**, but rather to address a practically important and underexplored setting: **zero-shot jailbreak defense without relying on supervised jailbreak examples on safety steering**. In this sense, “beyond the support” specifically refers to remove reliance on **Supervised training on jailbreak samples** , which is the main limitation of prior supervised steering methods.
> > >
> > > Our key contribution is a **bi-level adversarial training framework that simulates diverse out-of-support jailbroken activations in latent space**, enabling the steering field to learn from a substantially expanded attack activation region, even in the absence of real jailbreak prompts.
> > >
> > > For the subspace coverage analysis, we agree this is a proxy rather than a formal causal explanation. Our intention is to use it as an **interpretive indicator for simulated attack-space expansion**, rather than as a complete robustness theory.
> > >
> > > We sincerely appreciate your feedback, which helps us better position the paper around its central contribution: **zero-shot jailbreak defense for safety steering.**.

---

### Decision · Program_Chairs · 2026-04-30

**Decision:**

Accept (regular)

**Comment:**

The paper addresses an important problem in LLM safety: improving the generalization of activation-steering defenses beyond a fixed supervised set of jailbreak attacks. To do so, it proposes a bi-level adversarial training framework in which the inner step simulates diverse jailbreak activations by extrapolating from harmful-request refusal states via unsupervised latent direction discovery, while the outer step learns a potential-induced steering field that pushes such adversarial states toward refusal regions while preserving benign behavior. The reviewers found this overall formulation clear, timely, and conceptually novel. The empirical results across multiple models and jailbreak families were generally seen as strong, and the mechanistic analyses, including the subspace-coverage discussion, were considered a useful addition.

The reviewers have also raised several concerns. In particular, several reviewers noted that the paper’s generalization claims would be stronger with evaluation on more challenging or truly unseen attack settings, such as adaptive, multilingual, encoded, or more heavily obfuscated jailbreaks. There were also requests for clearer sensitivity analysis and more discussion of practical deployment issues such as inference overhead, long-generation stability, and the assumptions underlying latent direction discovery. The rebuttal has addressed the reviewers' concerns.

Overall, I find this to be a technically solid and well-motivated contribution. The paper presents a meaningful new perspective on zero-shot jailbreak defense, supports it with strong empirical results, and provides enough analysis to make the contribution useful to the community.

**Note**: one reference is detected as AI-generated. Please be careful in the final version.

Reference: Askell, A., Askell, G., Hadfield-Menell, D., et al. Constitutional ai: A framework for ethical ai development. arXiv preprint arXiv:2310.07590, 2023.
Issue: authors+title mismatch with arXiv